# Liquid Biopsy Approach for Pancreatic Ductal Adenocarcinoma

**DOI:** 10.3390/cancers11060852

**Published:** 2019-06-19

**Authors:** Etienne Buscail, Charlotte Maulat, Fabrice Muscari, Laurence Chiche, Pierre Cordelier, Sandrine Dabernat, Catherine Alix-Panabières, Louis Buscail

**Affiliations:** 1INSERM U1035, Bordeaux University, 33000 Bordeaux, France; ebuscail@me.com (E.B.); laurence.chiche@chu-bordeaux.fr (L.C.); sandrine.dabernat@chu-bordeaux.fr (S.D.); 2Department of Digestive Surgery, Bordeaux University Hospital, 33600 Pessac, France; 3Université Fédérale Toulouse Midi-Pyrénées, Université Toulouse III Paul Sabatier, INSERM, CRCT, 31330 Toulouse, France; charlotte.maulat@gmail.com (C.M.); muscari.f@chu-toulouse.fr (F.M.); pierre.cordelier@inserm.fr (P.C.); 4Department of Digestive Surgery, Toulouse University Hospital, 31059 Toulouse, France; 5Laboratory of Rare Human Circulating Cells (LCCRH), Montpellier Hospital and University of Montpellier, 34295 Montpellier, France; c-panabieres@chu-montpellier.fr; 6Department of Gastroenterology and Pancreatology, Toulouse University Hospital, 31059 Toulouse, France

**Keywords:** pancreatic ductal adenocarcinoma, circulating tumour cells, circulating cell free tumour DNA, extracellular vesicles, exosomes, KRAS oncogene, liquid biopsy

## Abstract

Pancreatic cancer is a public health problem because of its increasing incidence, the absence of early diagnostic tools, and its aggressiveness. Despite recent progress in chemotherapy, the 5-year survival rate remains below 5%. Liquid biopsies are of particular interest from a clinical point of view because they are non-invasive biomarkers released by primary tumours and metastases, remotely reflecting disease burden. Pilot studies have been conducted in pancreatic cancer patients evaluating the detection of circulating tumour cells, cell-free circulating tumour DNA, exosomes, and tumour-educated platelets. There is heterogeneity between the methods used to isolate circulating tumour elements as well as the targets used for their identification. Performances for the diagnosis of pancreatic cancer vary depending of the technique but also the stage of the disease: 30–50% of resectable tumours are positive and 50–100% are positive in locally advanced and/or metastatic cases. A significant prognostic value is demonstrated in 50–70% of clinical studies, irrespective of the type of liquid biopsy. Large prospective studies of homogeneous cohorts of patients are lacking. One way to improve diagnostic and prognostic performances would be to use a combined technological approach for the detection of circulating tumour cells, exosomes, and DNA.

## 1. Introduction

Pancreatic ductal adenocarcinoma (PDAC) cancer is a significant public health problem and a medical and scientific challenge because of its increasing incidence, the lack of reliable early biomarkers, and the absence of preventive screening and efficient therapies to overcome these aggressive and highly heterogeneous neoplasms [1,2,3]. The only curative treatment is surgery, which is only feasible in 15% of cases. For other patients, the tumour is already metastatic or locally advanced at the time of diagnosis. Despite application of chemotherapy protocols such as FOLFIRINOX (association of 5-fluoro-uracile, oxaliplatin and irinotecan), which increases survival after palliative or adjuvant therapies, the 5-year survival rate remains below 5% [4,5,6]. In parallel with the search for new treatments, several challenges must be faced to help alleviate the dismal prognosis of PDAC. One of these is to discover new biomarkers to ensure early detection, make a meaningful prognosis, and help predict recurrences. Ideally, these questions should all be addressed remotely, without the need for invasive, painful, and risky procedures. In this context, real-time liquid biopsy is an emerging tool of particular interest from the scientific and clinical perspective because complementary circulating biomarkers are released by the tumour and its metastases and therefore distantly reflect the disease [7,8,9,10]. Liquid biopsies are collected in a non-invasive manner and allow diagnosis and molecular follow-up of patients. They have already been implemented in the clinic to monitor and manage the therapies of a number of epithelial cancer patients [11,12].

The purpose of this review is to address all molecular, technological and clinical aspects and issues of liquid biopsies, as they relate to PDAC. Circulating tumour cells (CTCs), cell-free circulating tumour DNA (ctDNA), circulating tumour extracellular vesicles (e.g., exosomes), tumour educated platelets (TEPs), and blood-based protein and metabolite markers are specifically discussed, because they provide pragmatic perspectives on the application of promising technologies to the manage patients at all stages of PDAC.

## 2. Current Diagnosis for Pancreatic Cancer

PDAC diagnosis usually relies on information obtained using sequential procedures, including imaging data (ultrasonography, computerised tomodensitometry (CT), magnetic resonance imaging (MRI) and endoscopic ultrasound (EUS)). Blood markers are lacking, and the only one used is the carbohydrate antigen 19-9 (CA 19-9) which displays too low sensitivity and specificity to be useful as a stringent diagnostic marker for PDAC. Although CA19-9 is an acceptable marker for advanced and symptomatic tumours (sensitivity of 80%; specificity of 82%), its performance falls rapidly in the case of small non-metastatic lesions [13]. The different imaging techniques aim to detect a pancreatic tissue mass and to determine its spread locally by evaluating possible venous and arterial vascular invasion, but also at distance to detect metastases, specifically peritoneal and hepatic metastases. These are sometimes detected by diffusion MRI. After having assessed its spread, the tumour will be classified as either locally advanced and non-resectable (25% of patients at the time of diagnosis), metastatic (50%), borderline (10%), or resectable (15%). The performance of the CT and the MRI are generally equivalent for the diagnosis and assessment of pancreatic cancer staging [14], with CT more effective for the diagnosis of tumour unresectability. However, in the majority of cases (except for resectable tumours) a pathological confirmation is required either with a fine-needle aspirate of a metastasis or a EUS-guided fine-needle aspiration (EUS-FNA) of the primary tumour. More than 20 years of EUS experience have now paved the way for safe guided FNA biopsies of solid pancreatic lesions for cytopathological analysis [3,15]. EUS-FNA is thus now an effective technique to diagnose and assess the staging of PDAC, especially for tumours less than 2 cm [16,17,18]. However, its performance (and performances of other imaging techniques) greatly depends upon the operators’ experience and certainly on the nature of the PDAC, depending on the importance of stroma. In other words, the more abundant the tumour stroma, the fewer carcinoma cells will be present in EUS-guided microbiopsy samples. Accuracy of EUS-FNA to diagnose malignancy varies widely, with sensitivity ranging from 65% to 95% with a mean accuracy of 85%. In addition, the negative predictive value still ranges from 50% to 70% and the EUS-FNA may be inconclusive or ambiguous in up to 20% of cases [16,19,20] of PDAC. Inconclusive specimens can be defined as the presence of coagulum with normal cells or acellular samples. Ambiguous samples can be defined by the presence of atypia and/or low-grade dysplasia and/or be atypical for the malignancy. Recently, technical and clinical improvements to EUS have been developed such as elastography, contrast-enhanced EUS, and the use of an on-site pathologist. Each of these procedures may improve EUS for tissue characterizations [14,21] as well as for differentiating between PDAC and neuroendocrine tumours, lymphoma, or autoimmune pancreatitis [16,17]. Nevertheless, in cases of strongly suspected PDAC, a negative EUS-FNA result remains problematic. A second EUS-FNA is therefore required, equating to a treatment delay for the patient, which is known to negatively influence the prognosis of PDAC [22]. In addition, there is the problem of the differential diagnosis between PDAC and chronic pancreatitis in its pseudo-tumoural form and, more rarely, with autoimmune pancreatitis. In these cases, histological diagnosis is also crucial to avoid unnecessary surgery.

To assist in cytopathological diagnosis, the research and characterisation of new molecular markers is an active field. Nevertheless, since the oncogenic KRAS point mutation is a frequent event during PDAC, the identification of this mutation in tumour tissues may aid diagnosis. We and others have demonstrated that KRAS-mutation analysis, performed on EUS-FNA samples, appears to be highly accurate in differentiating benign versus malignant pancreatic solid lesions [19,20,23]. Faced with solid pancreatic masses, combining results of the KRAS-mutation assay with cytopathology can increase the sensitivity and accuracy of diagnoses when compared to cytopathology alone [20,24,25,26]. More importantly the negative predictive value of cytopathology alone for this indication is increased from 67% to 88% when combined with a KRAS mutation assay [20,24,25,26]. KRAS mutation detection has also been performed on circulating cell free tumour DNA and on isolated exosomes. The hope is to obtain a non-invasive, reliable, and reproducible biological marker demonstrating the presence of PDAC, irrespective of stage. We will discuss the potential of the different liquid biopsy approaches applicable to PDAC by trying to compare performance, weaknesses, and prospects.

## 3. Circulating Tumour Cell-Based Diagnosis of Pancreatic Cancer

The clinical utility of CTC detection for cancer patients regained interest during the late 1990s, and is now referred to as liquid biopsy [10]. Progress with regards to CTC capture/identification has predominantly relied on technical advances. Numerous CTC detection methods have been described. These primarily rely on a first step of CTC capture/enrichment based on tumour cell biological (immunological positive or negative selection) or physical properties (size, deformability, cell density), followed by a second detection step based on immunocytochemistry, molecular biology, or a functional assay [27]. Liquid biopsies can be performed on several body fluids, but application to pancreatic cancer detection has mainly been performed on total peripheral blood [28].

At first, liquid biopsies relating to pancreatic cancer used methods combining density centrifugation and RT-PCR detection of tumour markers, which were already known to have a poor specificity as circulating tumour biomarkers. This is exemplified by the low detection rates with CEA mRNA detection (26%, [29]), cytokeratine 20 (CK20, 34%, [30]), and EpCAM (epithelial cell adhesion molecule) (25%, [31]). These low detection rates could not be overcome by recruiting a higher proportion of early stage patients [29,30], except in the case of EpCAM RNA-based detection, since most patients were eligible for up-front surgery (83%, [31]). With increasingly available tools allowing the specific capture of epithelial cells from whole blood, numerous studies have tested combined immunocytochemistry detection of tumour markers on isolated cells. First, the CellSearch© system, currently considered as the gold standard because it has been cleared by the U.S. FDA (Food and drug Administration) for use in metastatic breast, colon, and prostate cancer, has been tested in several studies. This system allowed a PDAC diagnosis to be made in 11–48% of patients in cohorts comprising at least 53% of patients with locally advanced or metastatic disease (Table 1) [29,30,31,32,33,34,35,36,37,38,39,40,41,42,43,44,45]. Even when all the patients in the cohort had advanced disease, rates of CTC detection were not observed to increase (with one result for example being 32.3% [39]). As expected, when 100% of the patient cohort comprised resectable cases, the number dropped to less than 7% [42]. These quite disappointing results were usually explained by the fact that the CellSearch^®^ system is based on EpCAM and cytokeratin expression. These epithelial markers are down regulated and might even be lost during the epithelial–mesenchymal transition (EMT, [46,47]), which is a well-accepted fact for CTCs [48]. To limit this detection bias, CTC enrichment of blood samples has been tested using CTC physical properties. The isolation by size of epithelial tumour cells method (ISET, [49]) might be a good alternative for CTC detection, as it is unbiased by the needed presence of cell surface EpCAM. After whole blood cell filtration on a microporous membrane, retained cells are stained by routine histology with or without pan-cytokeratin immunocytochemistry. CTC identification and counting is performed by pathologists trained to identify the specific morphology of tumour cells [49]. In a cohort comprising more than 80% metastatic patients, for which results using both methods were available for 27 patients, the detection rate reached 93% for ISET as compared to only 40% for CellSearch^®^ with higher mean numbers of CTCs (6 vs. 26 CTC/7.5 mL, Table 1). Another cohort including mainly patients with early stage disease found 78% of PDAC [37], even when pan-cytokeratin positivity was associated with CTC detection. The ScreenCell^®^ filtration method identified 67% of PDAC patients, including 72% of advanced diseases, with cytomorphologic criteria after filtration [38,43]. The latter study included the detection of mutant KRAS in CTCs, with high discrepancy between tumour and CTC status. Indeed, whereas 97% of tumours carried mutant KRAS, 18% of the CTCs were found to carry only the KRAS wild type allele. Even CTCs from five out of the 12 metastatic tumours were KRAS WT (Wild Type). Thus, it is possible that cytomorphological-based CTC identification of filtered cells might falsely consider epithelioid or endothelial cells as epithelial tumour cells, leading to a general CTC overestimation. This hypothesis is supported by a report analysing 171 blood samples from patients with various pancreatic diseases (including 63% PDAC, [50]) and nine healthy controls. The nine healthy controls were free of circulating epithelioid cells (CEC), but of the 115 patients with CECs, 25 (15%) had non-malignant diseases. Morphologic characteristics of malignant CECs were undistinguishable from non-malignant CECs. In addition, CECs were also detected in inflammatory benign colonic diseases [51], suggesting that in specific cases only, these methods might detect cells shed by primary tumours as well as by benign lesions. We may also argue that identification of the KRAS mutation, which is detected very early during PDAC oncogenesis, may help detect early disease, which is eligible for surgery, before tumours grow too big or even disseminate.

Most studies nevertheless correlate the presence and/or number of CTCs with clinical parameters (Table 1). As expected, high CTC numbers correlated with metastatic disease [34] with a worse overall or progression free survival [37,38,39,41,42,45]. When sensitivity of detection is low (11%), CTC positivity correlates with adenocarcinoma differentiation [52].

Overall CTC-based diagnostic of PDAC is highly specific, since most studies with healthy control groups report close to 0 false-positive results (except for 3.6% in [35]). However, sensitivity suffers from the rarity of CTCs which are not efficiently captured/enriched by currently available methods. Filtration and morphologic-based CTC identification carries the risk of over interpretation [50]. In summary, the presence of CTCs from resected, locally advanced, or metastatic tumours was associated with poor PDAC patient prognoses, an increased number of metastases and an increased likelihood of relapse. As shown in Table 1, 13 out of 18 studies concluded that the presence of CTCs indicate poor survival. This is also illustrated in Figure 1.

## 4. Circulating Tumour DNA for Diagnosis and Prognosis of Pancreatic Cancer

ctDNA originates from necrotic or apoptotic cells but can also be actively secreted by cells. CtDNA is highly fragmented with a median size of 170 base pairs, which corresponds to internucleosomal DNA fragments. CtDNA is cell-free, circulating tumour DNA, because it contains mutations that are only specific to cancer cells [9,53]. Only very few studies have investigated methylation, microsatellite instability and allelic imbalance in ctDNA [7,54]. In contrast, the detection of KRAS mutation in plasma and serum appears the most widely used approach [7,55,56,57,58,59,60,61,62,63,64,65,66,67,68,69,70,71,72,73,74,75,76,77,78]. Various effective methods have been developed for KRAS mutation analysis and now replace direct sequencing and other methods requiring a pre-amplification step, such as RFLP. All these new methods include q-PCR methods, allele-specific PCR using amplification refractory mutation system technology or co-amplification at a lower denaturation temperature, PCR methods, pyrosequencing approaches, and real-time PCR methods that use specific probe technologies, such as peptide nucleic acids [62,79,80,81,82]. Digital droplet PCR (dPCR) is another approach which displays exceptional sensitivity and only requires a low starting concentration of DNA template. For example, considering the sensitivity of different KRAS mutation detection techniques (i.e., ratio mutant/wild type KRAS) that of direct sequencing is 10–30%, while that of next generation sequencing (NGS) is 10% and that of dPCR is 0.01% [83,84].

Numerous studies have been conducted to investigate the role of circulating DNA in patients with PDAC. The major studies that included more than 20 patients are compiled in Table 2 [55,56,57,58,59,60,61,62,63,64,65,66,67,68,69,70,71,72,73,74,75,76,77,85]. Of the 24 studies, only 13 included patients with benign pancreatic lesions or healthy subjects. Most studies have applied the detection of the KRAS oncogene mutation to identify circulating tumour DNA. Two questions arise with regard to the detection of circulating DNA: Does it have a diagnostic advantage (and/or can it detect early lesions)? Does it have a prognostic value?

It seems that the detection of KRAS mutations from blood still has limited value for the identification of early tumours or micrometastatic disease, which suggests that either a limited amount of ctDNA is released at these stages of the disease, or that the ctDNA concentration is so low and it is so degraded, that its detection requires more sensitive nucleic acid processing and analysis technologies. In the specific context of the PDAC diagnosis, despite the use of the most efficient techniques such as dPCR, there is the problem of sensitivity of the KRAS assay. Indeed, concordance studies have been carried out between the presence of the KRAS mutation in the primary tumour and the search for the mutation in ctDNA: the concordance varies from 25% to 75% and finally the sensitivity of this approach is highly dependent on the nature of the tumour [7,54]. If we analyse the results displayed in Table 2, the presence of a KRAS mutation in ctDNA is observed in nearly 70–80% of locally advanced and metastatic patients while this value ranges from 30% to 68% for patients with resectable tumours. One simple explanation is that the quantity of ctDNA might depend on the number of PDAC cells in the blood as well as the extent of metastasis. However, this postulate remains to be proven. In addition, and as discussed above, the sensitivity is improved in the case of dPCR (43–78%) when compared to simple PCR or sequencing (27–47%) (Table 2).

Several groups, including ours, have investigated whether the presence or absence of a KRAS mutation can influence the prognosis of PDAC [68,86,87,88,89,90]. Biological samples are varied and include tumour tissues, blood, plasma, or EUS-FNA. Overall and regardless of the type of biological sample used, the presence of a KRAS mutation has a negative influence on the prognosis of PDAC patients whether or not they undergo surgery (with complete tumour resection or whether they have locally advanced and/or metastatic PDAC). As shown in Table 2, 17 out of 24 studies concluded that the presence of a KRAS mutation had an adverse effect on survival (Figure 1).

Some studies also pointed out that the KRAS mutational subtype might also negatively influence prognosis per se (such as G12D and G12V) [38,64,68]. A different coupling to the downstream-signalling pathways of the KRAS protein, depending on the type of mutation, may explain these results [91,92,93].

On the whole, it is highly likely that ctDNAs will play a role in PDAC prognosis. However, new prospective studies are required, which specifically include control patient groups (i.e., free of pancreatic disease and not suffering from chronic pancreatitis) in order to clearly evaluate the sensitivity, specificity and predictive values of this assay for a possible non-invasive diagnosis of PDAC. One more issue is the variability of the detection assay itself in the absence of “universal” threshold and quantification values. To gain further insight and reach a definitive conclusion, multicentre studies in a larger (i.e., more than 200 patients), homogeneous cohort of patients (to allow strong multivariate analyses) are certainly needed. These studies must include sequential sampling of blood in order to establish the real reproducibility of the methods and to evaluate the potential performance, to either predict the response to treatment or the likelihood of relapse. The fragmentation partner of ctDNA itself can help distinguish healthy donor from patients with cancer, including pancreatic adenocarcinoma [94]. Indeed, profiles of healthy individuals reflected nucleosomal patterns of white blood cells, whereas patients with cancer have largely altered fragmentation profiles. Using artificial intelligence, researchers were able to detect breast, colorectal, lung, ovarian, pancreatic, gastric or bile duct cancer inpatients ranging from 57% to more than 99% among the seven cancer types at 98% specificity, with an overall area under the curve value of 0.94. Fragmentation profiles could also be used to identify the tissue of origin of the neoplasm. Combining fragmentation profiles with mutation-based cell-free DNA analyses detected 91% of patients with cancer.

Beyond mutational burden, methylation of ctDNA has recently emerged as a promising approach for cancer risk assessment and monitoring, especially for the detection of early tumours, including pancreatic cancer [95], or in colon cancer where a five-gene methylation panel can be used to compensate for the absence of patient-specific mutations to monitor tumour burden [96]. While challenging, this strategy has recently benefited from major technological advances [97], and could be proposed soon for assessing pharmacodynamics in clinical trials or when conventional ctDNA detection or imaging prove to be of limited use.

## 5. Exosome-Based Diagnostic for Pancreatic Cancer

Circulating tumour extracellular vesicles (EVs), including exosomes, are enriched with many bioactive molecules such as RNA, DNA, proteins, lipids and metabolites [98,99]. Once released from parental healthy or cancer cells, the cargoes reflect the status of that parent cell, and have the ability to relay signals between cells. For example, pancreatic cancer-derived exosomes loaded with tetraspanin 8 recruit proteins and mRNA cargo that can activate angiogenesis-related gene expression in neighbouring non-tumour endothelial cells [100]. EVs can also act at a distance. For example, Kupffer cells, the resident macrophages in the liver, have been shown to uptake pancreatic cancer derived exosomes containing the macrophage inhibitory factor (MIF). MIF promotes TGF-β (transforming growth factor beta) secretion by Kupffer cells, which in turns stimulates neighbouring hepatic stellate cells to secrete fibronectin, and creates local inflammation, considered to be the niche of pancreatic metastases [101]. Thus, due to their actual demonstrated activity, the detection of transported biomolecules protected from degradation by external nucleases or proteases presents an opportunity for both the diagnosis and prognosis of cancer.

There is currently no universal method for EV isolation/enrichment from body fluids, although recommendations from the International Society for Extracellular Vesicles have been first listed in 2014, and updated in 2018 [102]. Focusing on studies interested in PDAC, several methods have been described, all of them being complemented by PDAC-specific molecular characterisation of the enriched samples. For instance, enrichment by ultracentrifugation with or without sucrose gradients and ultrafiltration, or by kit-based precipitation have been reported (Table 3). More specifically, antigen-based exosome capture is also available using the CD63 exosome-specific tetraspanin in conjunction with microfluidic systems or magnetic beads. As EVs are released by any healthy or diseased cell, additional molecular characterisation of the obtained vesicles is required. Authors have been focusing on bioactive molecules carried by EVs such as nucleic acids and proteins.

As for other types of cancers [103] microRNA (miR) identification has been studied in the context of pancreatic cancer. As presented in Table 3, several, distinct miR signatures have been reported. From the testing of four individuals miRs (miR-17-5p, -21, -155, and -196a), miR-17-5p and -21 were shown to have a high diagnostic value, with a sensitivity and specificity between 72% and 95%. miR-155 and -196a were disregarded due to their low levels of expression in cancer exosomes [104]. Conversely, Xu et al. described an increased abundance of miR-196a, miR196b, or miR1246 exosomes in PDAC patients with areas under ROC curves (AUCs) ranking between 0.71 and 0.81. Other authors have found an increase in the expression of miR-191, miR-451a, and miR-21 in pancreatic cancer and IPMNs. Diagnostic accuracy was better than with CA-19.9 for early stage cancers, at approximately 80% [105]. Using a microarray approach on patient exosome samples, miR-1246, -4644, -3976, and -4306 were each found to be increased in PDAC samples [106]. Although theses authors did not report individual specificities and sensitivities, which were not 100% according to published figures, they did report that when all four miRs were combined they were detected in 9% of healthy controls (false positives) and in 80% of PDACs (20% of false negatives).

The prognostic value of miR quantification in exosomes was not evaluated by all authors. Unlike miR-21, the expression of miR-17-5p was correlated with tumour stages [104], although Goto et al. did find miR-21 to be prognostic for overall survival and chemo-resistance [105]. Interestingly, miR-451a was associated with patients with mural nodules in IPMNs [105], which is a sign of malignancy [114]. Quantifying miR-451a in exosome liquid biopsies could help in decision making for surgery of branch duct IPMNs. On the whole, a prognostic value has been found in five out of the 11 studies detailed in Table 3 (Figure 1).

It should be noted that circulating miR detection has superior sensitivity compared to ctDNA in the surgical setting; indeed, pre-operative plasma miR-21 was recently found as a sensitive biomarker and independent prognostic factor in patients with pancreatic cancer undergoing surgical resection [115]. Our group has participated in the demonstration that circulating miR sampled from different sources have biomarker value in preclinical models of PDAC and in patients. Briefly, we generated the first signature of salivary miR sampled from patients with locally advanced pancreatic tumours and found that selected salivary miRs, among them miR-23a, miR-23b, and miR-21, differentiate pancreatic cancer patients from patients with pancreatitis and matching healthy controls [115,116]. Interestingly, in mouse models of pancreatic cancer, the increase in salivary miR-23a, miR-23b and miR-21 levels precedes tumour burden detection by imaging [116]. In addition, in patients treated by gene therapy, we found that a panel of plasma miRs is predictive of response to therapy [117]. Since then, our group has partnered with physicists to develop novel nanodevices for the detection and quantification of candidate miRs from patients [118].

MiRs are not the only nucleic acid cargo of the exosomes. Unlike ctDNA, whose maximal size ranges from 150 to 170 base pairs [119], exosomes contain >10 kb fragments of double-stranded genomic DNA with detectable KRAS and p53 mutations when obtained from PDAC patients [120]. The difference in length observed between ctDNA and exoDNA is explained by the fact that DNA in exosomes is protected from nucleases. The relevance of exoDNA is actually a recent concept, and only a few studies have compared ctDNA and exoDNA diagnostic performances. Of note, specificity seemed to be impaired by the strength of the digital PCR-based positive signal in the detection of mutant KRAS in exoDNA in non-neoplastic patients examined in two recent reports. With mutation detection rates of 25% (3/12, [79]) and 7.4% (4/54, [109]) reported respectively. The KRAS mutation has already been described in a non-negligible proportion of plasma from healthy individuals (14/394, 3.5%, [121]) possibly reflecting a rare spontaneous somatic mutations. As for ctDNA, the high rates obtained in exoDNA might be linked to the highly sensitive detection method of KRAS mutants, which necessitate an interpretation considering the mutant allele frequency (MAF, [79]). The diagnostic performance of exoDNA was somewhat disappointing, since diagnostic accuracies ranked between 35% and 69% [79,109]. However, exoDNA showed relevance in PDAC management, as a correlation with progression free survival was detected in both studies, but this correlation was limited to patients with metastatic disease [106,113]. Despite its good prognostic value, exoDNA based on mutant KRAS detection using highly sensitive detection techniques, might not be suitable on its own for general population screening as it yields a high rate of false positives [79,109]. Additional biomarkers such as miRs or proteins should be included in the screening plan to improve the specificity.

Since protein cargo of EVs carry their biological function, diagnostic accuracy might be increased by protein marker identification. The membrane protein heparan sulfate proteoglycan glypican 1(GPC1) is overexpressed in several types of cancers, in particular in pancreatic primary tumours [122]. In 2015, a promising study quantified GPC1-positive exosomes in peripheral blood to diagnose PDAC and reported a sensitivity and a specificity of 100% [107]. Moreover, this circulating biomarker had strong prognostic value and could detect pre-neoplastic stages. In later studies, exosomal GPC1 alone failed to diagnose PDAC [108]. A signature with five exosomal surface proteins including GPC1 showed better sensitivity and specificity than each marker analysed separately (86% and 81%, [111]). Using an alternating current, electrokinetic (ACE) microarray chip that captures nano-particles including exosomes directly from plasma, followed by quantification of CD63 and GPC1, retrieved good but not perfect performance (sensitivity 99% specificity 82%) with false negatives and false positives [112]. Pulling down GPC1-expressing exosome did not increase diagnostic performance of miR-196a and miR-1246 [110]. Discrepancies in exosomal GPC1 validity in detecting PDAC might come from methodological differences. However, details on flow cytometry controls and settings published by Melo et al. raise questions on the validity of the published results. This might explain why no independent study has been able to confirm their results by using the same method. In addition to GPC1, the zinc transporter protein ZIP4 has recently been studied as an exosomal protein biomarker. The authors did not report diagnostic accuracy, but the AUC was 0.89. As was the case for GPC1, exosomal ZIP4 led to false-positive and false-negative results [113].

In conclusion, recent experimental applications utilising miRs, DNA, and exosome protein signatures, have not yet found a routine application for these techniques in the diagnosis of pancreatic cancer. Exosome isolation and characterisation is a major challenge in the field and is required to improve and standardise methods [102]. Another obstacle is the choice of the molecular signature. Exosomes are for instance very rich in differentially expressed miRs, as 119 miRs, with a 5-fold higher expression level detected in pancreatic cancer compared to healthy controls [106,123,124]. Among the numerous targets in each category, which have been selected by diverse strategies, only a few have been tested to date. Of the cited references here, it appears that combining miR-1246, miR-21, GPC1 and ZIP4 might be very relevant for the diagnosis of PDAC in particular.

## 6. Tumour-Educated Platelets

The team of Würdinger et al. has for many years explored how platelet RNA biomarker signatures can be altered by the presence of cancer [125] and reported an important role of these tumour-educated platelets (TEPs) as liquid biopsies in solid tumours (e.g., glioblastoma, non-small cell lung cancer, colorectal cancer, pancreatic cancer, breast cancer, and liver and bile duct carcinoma) [126]. Indeed, it has been previously shown that platelets interact with tumour cells and affect tumour growth and dissemination [127]. This interaction affects not only the expression of relevant genes in tumour cells, but also alters the RNA profile of blood platelets called TEPs. Interestingly, tumour-associated biomolecules can be transferred to platelets resulting in their education [125]. External stimuli (e.g., activation of platelet surface receptors and lipopolysaccharide-mediated platelet activation) induce specific splicing of pre-mRNAs in TEPs. Thus, RNA sequencing (thromboSeq technique) performed on 228 blood platelet samples from patients with different tumour types including pancreatic cancer showed that the location of the primary tumour was correctly identified with 71% accuracy [126]. Moreover, TEPs mRNA profiles could distinguish mutant KRAS, EGFR, or PIK3CA tumours, which is a crucial application in oncology. In conclusion, the ability of TEPs to identify precisely the location of the primary tumour as well as its molecular composition opens a new avenue to use liquid biopsies for cancer diagnostic. An important step in the future will be to demonstrate the clinical utility of TEPs as liquid biopsy biosources in large prospective clinical trials.

## 7. Blood-Based Protein and Metabolite Markers

Recent studies have addressed the role of circulating proteins for the diagnosis and the prognosis of PDAC. In most of these studies, the candidate biomarker is compared with CA 19-9. Two studies investigated the diagnostic value of thrombospondin-2 (THBS2), which is a disulfide-linked homotrimetric glycoprotein that mediates cell-to-cell and cell-to-matrix interactions [128,129]. The levels of THBS2 were compared in patients with PDAC, as compared to patients with benign pancreatic diseases. While the diagnostic performance of the THBS2 (ELISA) plasma assay is similar to that of CA19-9 (area under curve of the ROC 0.73 to 0.94) (AUC), the combination of the two assays significantly improves these performances (AUC: 0.90 to 0.97). A team from Texas University MD Anderson Cancer Center recently demonstrated that tissue metalloproteinase inhibitor-1 (TIMP1) and the Leucine-rich alpha-2-glycoprotein 1 (LRG12) expression detection shows significant performance for the diagnosis of early stage PDAC (AUC: 0.95), when associated with CA 19-9 detection [130]. In a similar study, they found that 5 metabolites show interesting diagnostic performance, either alone (AUC 0.72–0.84), or in combination (0.82–0.95). Combining this metabolomics signature with TIMP1, LRG1, and CA 19-9 detection gives a remarkable AUC of 0.92 [131]. Another team studied the diagnostic value of a combination of nine metabolites associated with the CA 19-9 assay. Of these nine candidates, five were able to discriminate PDAC and chronic pancreatitis with diagnostic values up to 0.95 [132]. Finally, a recent study revealed that high circulating levels of inhibitory immune checkpoints (such as PDL-1, BTLA, BTN3A1, and BTN3A) correlate with shorter survival [133].

## 8. Conclusions and Future Directions

The aim of liquid biopsies is to increase the negative predictive value of the gold EUS-FNA cytopathology standard but also to ensure the rapid diagnosis of PDAC in order to facilitate its early management. Other aims are to hopefully identify tools and conditions for a better evaluation of the prognosis as well as to better monitor PDAC patients, before and after surgical resection and in addition to allow prediction of relapses. However, there are still some unknowns regarding the use of liquid biopsies in the current practice for PDAC. Indeed, there is an obvious heterogeneity of the methods, regarding either the type of biomarker source “cells, DNA or exosomes” and the molecular targets used to identify them. The idea would be to get combined information as an index or an algorithm to get a more precise picture of cancer progression (e.g., quantitative and qualitative). On the other hand, standardisation is required before (or after) establishing the real place of liquid biopsies in the management of PDAC. With regard to diagnosis, the results are more convincing in the case of locally advanced and/or metastatic cancers, considering the size of tumour mass and diffusion. Nevertheless, the improvement of detection methods (such as the dPCR for the detection of the KRAS oncogene mutation) and the combination of liquid biopsy types (example of ExoDNA) should allow us to obtain a diagnostic yield that fits the current clinical practice. Thus, more emphasis on technical validation is required, and projects such as the European CANCER-ID, European Liquid Biopsy Academy (ELBA) and European Liquid Biopsy Society (ELBS) networks or the U.S.-based BloodPAC have been initiated to meet this challenge. These consortia combine the expertise of academic and industry partners and will hopefully lead to the development of robust liquid biopsy assays and inform the design of the trials needed to prove the clinical utility of liquid biopsy testing. These societies and consortia attempt also to find an agreement on the best way to collect, store and handle this type of sampling and analysis in such a way as to not only unify but also centralize it at expert centres.

Beside standardization of the methods, more studies are needed including a larger number of patients. Indeed, from a statistical point of view, most published studies have included between 50 and 75 patients with PDAC and the number of controls is not higher. It is obvious that any robust assessment of the sensitivity and specificity of a diagnostic method requires a sufficient number of patients, taking into account the heterogeneity of the disease (i.e., metastatic, locally advanced, borderline). The same is true for prognosis, which is certainly appreciable for all types of patients by measuring overall survival, but the evaluation of progression-free survival or recurrence after surgery is just as important. For this purpose, subgroups with a sufficient number of patients must also be studied.

Recent translational discoveries pinpoint that PDAC tumours are highly heterogeneous, at the molecular and cellular levels that can account at least in part to PDAC extreme resistance to treatment. This heterogeneity reflects the difficulty not only of early diagnosis but also of selecting reliable biomarkers. There are many markers and this is found in the field of liquid biopsies. Indeed, the only example of exosomes attests to this variety and heterogeneity since proteins, DNA, miRs coexist. This reflects the value of combining several methods. Recent examples have been given by the isolation of circulating exosomes carrying ctDNA or proteins of interest [79,99].

The role of liquid biopsies to assess PDAC prognoses is also evident, but in the absence of a significant therapeutic arsenal, the current therapeutic attitude for locally advanced or metastatic cancers will continue to be used. However, this is not to imply that we should sit twiddling our thumbs; our efforts must be doubled to identify biomarkers but also nanodevices for detection in the likely event that progress will be made in the therapeutic management of these advanced forms of the disease.

On the other hand, for patients with a resectable tumours and/or without obvious metastases, the contribution of liquid biopsies for the diagnosis of small metastases or the prediction of an early relapse after surgery would be of great benefit. Indeed, the surgical decision, the neoadjuvant treatment, and the intensification of adjuvant treatment could all be modified and decided on this basis.

In addition, these new non-invasive methods which lend themselves to frequent sampling, could certainly benefit from a combination of several approaches by concomitantly detecting several circulating tumour elements. A representative example is the exosomes that are carriers of tumour DNA; these might indeed be representative of tumour spread and metastasis. This approach would allow the evaluation and monitoring of the “distant activity” of PDAC. In addition, the use of NGS or miR profiling containing specific “molecular signatures” that have been already characterised in pilot studies (as detailed in the present review) may also improve the sensitivity of liquid biopsies in PDAC and could be integrated as a tool for personalised medicine.

The improvement of technology for detection and isolation of CTCs and exosomes will certainly be of importance, as well as pilot studies comparing blood samples near the tumour (portal vein in the case of PDAC) and peripheral blood. In addition, isolation of CTCs will be also be a great model for in vivo studies of tumour progression and response to treatment.

In conclusion, all published studies suggest that real-time liquid biopsy is a highly promising clinical tool for the non-invasive diagnosis and prognosis of PDAC in the future. Combination of methods will certainly be the key point of these promising modern methods of diagnosis.

## Figures and Tables

**Figure 1 cancers-11-00852-f001:**
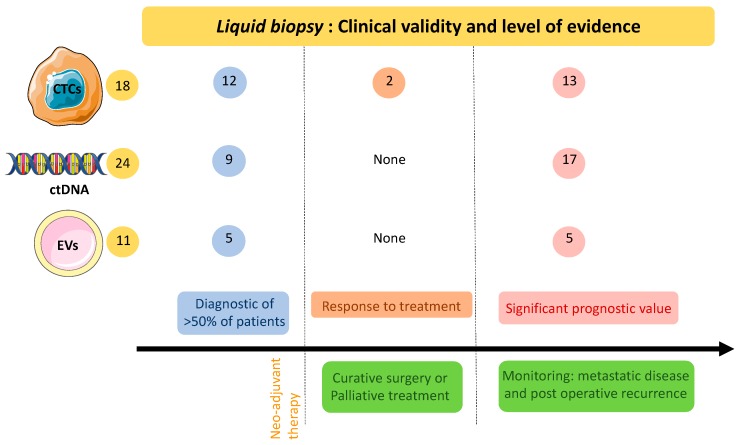
Clinical validity of circulating tumour elements in pancreatic cancer patients as reported in published major clinical studies. Yellow circles: absolute numbers of publications; Blue circles: number of publications with a significant correlation between diagnosis and the presence of circulating tumour elements; orange circles: publications with a significant correlation between response to neoadjuvant therapy and the presence of circulating tumour elements; light red circles: publications with a significant correlation between prognosis and the presence of circulating tumour elements. CTCs: circulating tumour cells; ctDNA: circulating tumour DNA; EVs: extracellular vesicles.

**Table 1 cancers-11-00852-t001:** Results of main clinical pilot studies assessing performances of circulating tumour cells (CTCs) detection in the diagnosis and prognosis of pancreatic ductal adenocarcinoma (PDAC).

PDAC Patient Number (Control)	Type of Tumour: Resected, Locally Advanced, Metastatic, All	CTC Enrichment	CTC Detection	CTC Count	CTC Detection Rate in PDAC Patients	Prognosis Value of CTCs	Reference
20 (15 benign diseases)	All (Samples before treatment)	Density centrifugation	RT-PCR CEA	NA	26%	Positive correlation with recurrence	Mataki et al., 2004 [29]
154 (68 benign diseases)	All (Samples before treatment)	Density centrifugation	RT-PCR CK20	NA	34%	Shorter OS (meta.) (*p *= 0.05)	Soeth et al., 2005 [30]
25 (15 benign diseases)	All (Samples before treatment)	Immunomagnetic (EpCAM)	RT-PCR: cMET, hTERT, CK20, CEA	NA	80–100% (sensitivity 100%; specificity 96%)	Not studied	Zhou et al., 2009 [32]
41 (20 HC)	All (Sample before and post treatment)	Immunomagnetic (leukocytes CD45^+^ depletion)	ICC: CK8/CK18^+^, CA19-9^+^, CD45	16	80% before and 20% after chemotherapy	Not studied	Ren et al., 2011 [33]
48 (10 CP)	All (Samples before and after treatment)	None	Real-time RT-PCR mRNA EpCAM	NA	25% pre-operative 65% post-operative	No correlation with any outcome	Sergeant et al., 2011 [31]
54 (No)	All (Sample time: NA)	Immunomagnetic: ISET and CellSearch^®^	ISET: Cytology, CD45^−^ ICC: CK^+^, DAPI^+^, CD45^−^	- ISET: 26 - CellSearch^®^:6	ISET:93% CellSearch^®^:40%	No correlation with any outcome	Khoja et al., 2012 [34]
79 (No)	LA (Samples before and after chemotherapy)	Immunomagnetic: CellSearch^®^	ICC: CK^+^, DAPI^+^, CD45^−^	1 to 15 (only 1 or 2 patients)	11%	Poor differentiation and shorter OS (*p* = 0.01)	Bidard et al., 2013 [52]
72 (28 benign diseases)	All (Samples before treatment)	Microfluidic (NanoVelcro)	ICC: CK^+^, DAPI^+^, CD45^−^ KRAS mutation	0 to ≥5 (*)	75%	≥3 CTCs: discriminate metastatic disease (*p* < 0.001)	Ankeny et al., 2016 [35]
48 (No)	Metastatic (Samples before treatment)	Immunomagnetic: CellSearch^®^	ICC: CK+, DAPI^+^, CD45^−^, MUC-1^+^	23 patients: ≥1 9 patients: ≥2	48%	CTC MUC-1^+^ correlate with a shorter OS (*p* = 0.044)	Dotan et al., 2016 [36]
60 (no)	All (40% of the samples performed after neo-adjuvant therapy)	Size based ISET	ICC: CK^+^, ALDH^+^, CD133^+^, CD44^+^	Mean: 7.1 Median: 4	78%	CK^+^/ALDH^+^: shorter OS and DFS CK^+^/CD133^+^/CD 44^+^: shorter DFS	Poruk et al., 2017 [37]
58 PDAC (10 HC)	All (samples time NA)	Size based: Screencell©	Cytology KRAS mutation	Range 0–13	67%	>3 CTC^+^: shorter OS	Kuleman et al., 2017 [38]
52 (10 benign diseases)	All (samples time NA)	Size based Screencell©	Cytology	Median 4 Range 0–151	67%	No correlation	Sefrioui et al., 2017 [43]
65 (15 HC)	LA and Meta. (Samples before treatment)	Immunomagnetic CellSearch^®^	ICC: CK^+^, DAPI^+^, CD45^−^	4.9	32.3%	Independent predictor of shorter OS	Okubo et al., 2017 [39]
100 (26 benign diseases)	All (32% of the samples after neo-adjuvant therapy)	Microfluidic Nano-velcro	ICC: CK^+^, DAPI^+^, CD45^−^	NA	78%	Correlated with presence of occult metastasis, shorter PFS and OS	Court et al., 2018 [40]
69 (9 benign diseases)	All (10% of the samples after neo-adjuvant therapy)	Immunomagnetic MACS and CellSearch^®^ (*n* = 20)	ICC: CK^+^, DAPI^+^, CD45^−^	17 patients >1 13 patients >2	33.3%	Independent predictor of shorter PFS and OS	Effenberger et al., 2018 [41]
242 (No)	All (sample time NA)	Immunomagnetic CellSearch^®^	ICC: CK^+^, DAPI^+^, CD45^−^	Median 1 Range 1–33	78.5%	Shorter PFS (*p* < 0.001)	Hugenschmidt et al., 2018 [42]
24 (no)	Metastatic (Samples before and after chemotherapy)	Microfluidic	ICC: CK+, DAPI+, CD133, EpCAM+, CD45-	Mean 3.87 CTCs/mL	84.4%	No correlation	Varillas et al., 2019 [44]
100 (16 benign disease, 30 HC)	All (Samples before and after treatment)	Microfluidic	ICC: Vimentin+, EpCAM+, CD45-	Median 3 Range 0–23	76%	≥2 CTCs vimentin+: correlate with a shorter PFS	Wei et al., 2019 [45]

CTCs: circulating tumour cells; LA: locally advanced PDAC; Metastatic: metastatic PDAC: All: resected + locally advanced + metastatic PDAC patients; NA: not available; HC: healthy control; CP: chronic pancreatitis; MACS: magnetic activation cell search; ISET: isolation by size of epithelial tumour cells; ICC: immuno-cyto-chemistry; DAPI: 4’, 6-diamidino-2-phénylindole as fluorescent protein linking to thymine and adenine DNA bases; OS: overall survival rate; PFS: progression free survival; DFS: disease free survival; NA: not available; MUC: mucin; EpCAM: epithelial cell adhesion molecule; ALDH: aldehyde dehydrogenase. CTC count is mostly expressed as mean/7.5 mL. (*): 18 patients: 0 CTC; 54 patients: ≥1 CTC; 39 patients: ≥2 CTCs; 29 patients: ≥3 CTCs; 18 patients: ≥5 CTCs.

**Table 2 cancers-11-00852-t002:** The main studies that have investigated the role of ctDNA in the diagnosis and/or prognosis of pancreatic ductal carcinoma.

PDAC Patient Number (Control)	Type of Tumour: Resected, Locally Advanced, Metastatic, All	Site	Target for ctDNA	% of Mutations or Genetic Alterations in PDAC Patients	Diagnosis Performances	Positive Correlation with a Poor Prognosis (OS) (*p*) *	Reference
44 (60: 37 CP and 23 miscellaneous)	All	Plasma	KRAS mutation Amplified PCR	27	Sensitivity: 27% Specificity: 100%	Yes—<0.005	Castells et al., 1999 [55]
47 (31: CP)	All	Serum	KRAS mutation sequencing	47	Sensitivity: 47% Specificity: 87%	No—Ns	Maire et al., 2002 [56]
56 (13: CP)	All	Plasma	KRAS mutation PNA-mediated PCR clamping and real-time PCR	36	Sensitivity: 36% Specificity: 100%	No—0.10	Däbritz et al., 2009 [57]
91 (No)	LA + Meta.	Plasma	KRAS mutation sequencing	33	-	Yes—<0.001	Chen et al., 2010 [58]
36 (49: 25 HC and 24 miscellaneous)	All	Plasma	KRAS mutation cold-PCR combined with an unlabelled-probe HRM	72	Sensitivity: 81% Specificity: 87.5%	-	Wu et al., 2014 [59]
27 (No)	LA + Meta.	Plasma	KRAS mutation ARMS PCR	37	-	Yes—0.003 Yes—0.014 ***	Semrad et al., 2015 [60]
51 (No)	R	Plasma	KRAS mutation dPCR	43	Sensitivity: 43% Specificity: >99%	Yes (predictor of disease recurrence)—0.015	Sausen et al., 2015 [61]
45 (No)	All	Plasma	KRAS mutation dPCR	26	-	Yes—0.001	Earl et al., 2015 [62]
110 (25: HC)	All	Plasma	KRAS mutation RFLP + sequ. Two-step enriched-nested PCR	31	-	No—0.36	Singh et al., 2015 [63]
75 (40: 20 CP and 20 HC)	All	Serum	KRAS mutation dPCR	63	-	Yes—0.024	Kinugasa et al., 2015 [64]
259 (No)	All	Plasma	KRAS mutation dPCR	8 (R), 18 (LA), 59 (M)	-	Yes—<0.0001	Takai et al., 2015 [65]
105 (20 HC)	R	Plasma	KRAS mutation dPCR	31	-	Yes—<0.0001 Yes—<0.001 **	Hadano et al., 2016 [71]
40 (10 HC)	All	Plasma and serum	KRAS mutation dPCR	48 (All) 38 (LA) 63 (LA and Meta. Serum)	-	Yes—<0.01	Ako et al., 2016 [67]
188 (No)	Met	Plasma	KRAS mutation dPCR	83	-	Yes—0.019	Cheng et al., 2017 [68]
135 (No)	All	Plasma	KRAS mutation NGS/dPCR	41 (LA and Meta.)	-	LA + Met: Yes—*p* < 0.001 Resected: Yes—0.027; Yes—0.03 **	Pietrasz et al., 2017 [69]
60 (No)	LA + Meta.	Plasma	KRAS mutation BEAMing	65	-	Yes—0.001 Yes—0.0022 **	Van Laethem et al., 2017 [70]
95 (No)	All	Plasma	28 genes Methylation-specific PCR	27 (>10 hypermethylated genes)	-	Yes	Henriksen et al., 2017 [85]
26 (26: 14 CP and 12 HC)	All	Plasma	KRAS mutation dPCR NGS: KRAS, SMAD4, CDKN2A and TP53	NGS: 27 dPCR: 23	-	Yes—0.018 ****	Adamoet al., 2017 [66]
27 (43 HC)	LA + Meta.	Plasma	KRAS mutation dPCR	70.4	-	No—0.16—0.24 ***	Del Re et al., 2017 [72]
221 (182 HC)	R	Plasma	KRAS mutation PCR Safe-Sequencing System	30	Sensitivity: 30% Specificity: 99.5%	-	Cohen et al., 2017 [73]
34 (No)	All	Plasma	NGS: 25 genes (including KRAS)	25 genes: 74 KRAS only: 29	-	Yes—0.045	Pishvaian et al., 2017 [74]
106 (No)	All	Plasma	KRAS mutation dPCR	68(R), 72(LA), 87(M)	Sensitivity: 78% Specificity: 33%	Yes—0.008 Yes—0.003 ***	Kim et al., 2018 [75]
65 (20 HC)	All	Plasma	KRAS mutation dPCR	80	-	No—0.73	Lin et al., 2018 [77]
45 (No)	R	Serum	KRAS mutation teal-time quantitative PCR	55	-	Pre-operative samples: No—0.258–0.710 ** Post-operative samples: Yes—0.027 **	Nakano et al., 2018 [76]

* A worse prognosis in patients with a mutated KRAS vs. wild type in term of overall survival (OS). **: disease-free survival. ***: progression-free survival. ****: disease specific survival. CP: chronic pancreatitis; HC: healthy controls; R: resected PDAC; LA: locally advanced PDAC; M or Meta.: metastatic PDAC: All: resected + locally advanced + metastatic PDAC patients; RFLP: restriction fragment length polymorphism; dPCR: digital droplet PCR; NGS: next generation sequencing; BEAM: Beads Emulsion Amplification Magnetic; ARMS PCR: amplification-refractory mutation system polymerase chain reaction; Cold-PCR: co-amplification at lower denaturation temperature-PCR; PNA-mediated PCR: peptide nucleic acid-mediated PCR; HRM: high resolution melt.

**Table 3 cancers-11-00852-t003:** Results of main clinical pilot studies assessing performances of exosome detection in PDAC.

Patient Number (PDAC)	Type of Tumour: Resected, Locally Advanced, Metastatic, All (Treatment)	Molecular Target(s)	Method of Isolation	Exosomes Detection Rate in PDAC Patients	Exosomes Diagnosis Performances	Exosomes Prognosis Value	Reference
16 (6 HP, 6 CP, 5 cysts, 5 ampullary carcinoma)	All (12 metastatic)	miR-17-5p, -21, -155	Ultracentrifugation RT qPCR	NA	(**)	miR-17-5p correlated with metastasis	Que et al., 2013 [104]
131 (64 HC)	All	miR-1246, -4644, -3976, -4306; CD44v6, TSPAN8, EpCAM, MET, CD104	Sucrose gradient, micro-array, RTqPCR, flow cytometry, latex beads	NA	Sens. 100% Spec. 80%	NS	Madhavan et al., 2014 [106]
146 (benign pancreatic diseases 32, 120 HC)	All (Neo-adjuvant: 10)	GPC1	Latex beads Ultracentrifugation Ac GPC1	100%	Sens. 100% Spec. 100%	GPC1+ correlates with worse DFS and OS	Melo et al., 2015 [107]
29 (CP 11)	Resected and locally advanced	GPC1 miR-10b, -21, -30c, -181a, -let7a	GPC1 LC-MS/ML RT qPCR	100%	Sens. 100% Spec. 100%	NS	Lai et al., 2017 [108]
127 (136 HC)	All	Exo DNA ctDNA	Ultracentrifugation Flowcytometry dPCR	54%	Sens. 54% Spec. 84% PPV 76% NPV 66%	Worse DFS *P* = 0.03 RR: 4.68 441 days vs. 127	Allenson et al., 2017 [109]
15 (15 HC)	All	miR: R196a, 196b and 1246	ExoKit RTqPCR NGS	Significantly higher for 196a and 1246	AUC: 196a: 0.81 1246: 0.73 196b 0.71	NS	Xu et al., 2017 [110]
68 (41 benign pancreatic diseases;18 HC)	All (Neo-adjuvant: 33)	Signature: EGRF, EpCAM, MUC1, GPC1, WNT2	Ultracentrifugation	89%	Sens. 86% Spec. 81%	NS	Yang et al., 2018 [111]
20 (20 benign diseases)	Resected and locally advanced	Protein CD63, GPC1	AC electrokinetics immunofluorescence	Significantly higher in PDAC cohort	Sens. 99 Spec. 82	NS	Lewis et al., 2018 [112]
32 (IPMN 29, 22 HC)	All	miR-191, -21, -451a	ExoKit Quick NGS RT qPCR		(*)	miR21 worse OS	Goto et al., 2018 [105]
24 (14 CP, 50 miscellaneous, 46 HC)	NA	Protein ZIP4	Exo Kit precipitation	Significantly higher in PDAC	AUC ROC curve 0.89	NS	Jin et al., 2018 [113]
194 (25 cysts, 12 HC)	All (123 metastatic)	Exo DNA KRAS	Ultracentrifugation ddPCR	61% metastatic 38% resectable	NS	MAF >5% Predictor PFS OS	Bernard et al., 2019 [79]

miR: microRNA; CP: chronic pancreatitis; HC: healthy control patients; IPMN: intraductal papillary mucinous neoplasia; NET: neuroendocrine tumour; NA: not available; NS: not studied; NGS: next generation sequencing; GPC1: sulfate proteoglycan 1; AUC: area under ROC curves; EpCAM; epithelial cell adhesion molecule; MUC1: mucin 1; TSPAN8: tetraspanin8; MAF: mutation allelic frequency. (*): miR-191: Sens. 71.9%, Spec. 84.2%, accuracy 76.6%; miR-21: Sens. 80.7%, Spec. 81%, accuracy 80.8%; miR-451a: Sens. 65.8%, Spec. 85.7%, accuracy 73.6%. (**): miR-17-5p: Sens. 72.7%, Spec. 92.6%; miR-21: Sens 95.5%, Spec. 81.5%.

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
