# Peer review of "Liquid Biopsy Approach for Pancreatic Ductal Adenocarcinoma"

_cancers, 2019, doi:10.3390/cancers11060852_

Round 1

Reviewer 1 Report

very interesting and good review

Author Response

The manuscript was initially proofread and edited by a proofreader whose native language is English from the translation company AmPLUS company, Lyon, France. The present version was further proofread and corrected by an English-speaking scientist. Numerous spelling and typo errors have been corrected throughout the text.

Reviewer 2 Report

The present manuscript gives an overview of the main recent clinical interest of circulating biomarkers in pancreatic cancers. The manuscript is well written and is in the scope of Cancers. My main concerns are:

- numerous works on CTCs and biomarkers in pancreatic cancers have been recently pusblished and not inserted in the manuscript. Several papers should be included and discussed, for instance (non exhaustive list):

- Microfluidic Isolation of Circulating Tumor Cells and Cancer Stem-Like Cells from Patients with Pancreatic Ductal Adenocarcinoma. Varillas JI, Zhang J, Chen K, Barnes II, Liu C, George TJ, Fan ZH. Theranostics. 2019 Feb 20;9(5):1417-142

- A Blood-Based Multi Marker Assay Supports the Differential Diagnosis of Early-Stage Pancreatic Cancer.Berger AW, Schwerdel D, Reinacher-Schick A, Uhl W, Algül H, Friess H, Janssen KP, König A, Ghadimi M, Gallmeier E, Bartsch DK, Geissler M, Staib L, Tannapfel A, Kleger A, Beutel A, Schulte LA, Kornmann M, Ettrich TJ, Seufferlein T. Theranostics. 2019 Feb 12;9(5):1280-1287.

- Vimentin-positive circulating tumor cells as a biomarker for diagnosis and treatment monitoring in patients with pancreatic cancer. Wei T, Zhang X, Zhang Q, Yang J, Chen Q, Wang J, Li X, Chen J, Ma T, Li G, Gao S, Lou J, Que R, Wang Y, Dang X, Zheng L, Liang T, Bai X. Cancer Lett. 2019 Jun 28;452:237-243.

- Prognostic significance of circulating PD-1, PD-L1, pan-BTN3As, BTN3A1 and BTLA in patients with pancreatic adenocarcinoma. Bian B, Fanale D, Dusetti N, Roque J, Pastor S, Chretien AS, Incorvaia L, Russo A, Olive D, Iovanna J. Oncoimmunology. 2019 Feb 3;8(4):e1561120.

- Detection of Solid Tumor Molecular Residual Disease (MRD) Using Circulating Tumor DNA (ctDNA). Chin RI, Chen K, Usmani A, Chua C, Harris PK, Binkley MS, Azad TD, Dudley JC, Chaudhuri AA. Mol Diagn Ther. 2019 Apr 2.

- Circulating Tumor Cells and Cell-Free DNA in Pancreatic Ductal Adenocarcinoma. Gall TMH, Belete S, Khanderia E, Frampton AE, Jiao LR. Am J Pathol. 2019 Jan;189(1):71-81.

- Pancreatic cancers are heterogeneous tumor at the genetic, epigenetic and transcriptomic levels. It should better discuss how circulating biomarkers may reflect the tumor heterogeneity and how such parameters may be used in the clinical practice.

Minor comments:

- The last sentence of ths abstract should be rephrased. Indeed, to detect various circulating tumor elements, several and diverse technological approaches are mandatory, there is nothing surprising. It would be more interesting to indicate the names of the various approaches.

- Ligne 360: What does "Bernard and XX respectively" mean?

- Ligne 362: replace "references 2 and 10" by 2,10 with brackets.

Author Response

- We fully agree that much works have been published on the interest of CTCs and other circulating tumour elements in pancreatic cancer. As far as possible, we selected works that included both patients with pancreatic cancer and patients with benign pancreatic disease or healthy controls, to address the diagnostic value of the approach. In addition, we selected only publications including more than 20 and 50 samples for Circulating Tumour Cells and ctDNA respectively (i.e. tables 1 and 2). Finally, we have given priority to studies questioning the prognostic impact of liquid biopsies.

Nevertheless, we agree that recent important studies were missing from this manuscript: they have been added to Table 1. Figure 1 has been corrected accordingly and references of these original articles and other reviews have also been added. In addition, we have added a chapter on circulating protein elements and markers (new chapter 7 and new references).

- Similarly, we have highlighted the important heterogeneity of pancreatic cancer, which is reflected at the level of liquid biopsies in terms of the multiplicity of methods, targets and impact on either the diagnosis or prognosis alone or sometimes both. On this occasion, we also highlight the interest of combining different liquid biopsy methods in order to increase the performance of this approach (a new paragraph in chapter 8 – lines 482-488).

All corrections and additional text appear highlighted in yellow.

- Minor comments:

The last sentence of the abstract has been rewritten as follow: “One way to improve diagnostic and prognostic performances would be to use a combined technological approach for the detection of both circulating tumour cells, exosomes and DNA.” 

Line 360: the two authors Bernard and Allenson have been verified and corrected/added with corresponding reference numbers

Line 362: references have been put in brackets

Reviewer 3 Report

The authors review the use of analyzing circulating tumor cells, circulating tumor DNA (more specifically mutations such as in KRAS), exosomes, and tumor-educated platelets as a means to diagnose PDAC. They conclude that (i) liquid biopsies are highly promising for diagnosis and prognosis, and that (ii) combination of diagnostic methods may hold the best promise to this end.

The manuscript consists of 8 pages of very dense text, one figure and three extended tables summarizing the carefully curated literature. The literature cited is fairly up to date with 25 out of the 120 references from 2018 or 2019.

Many reviews have previously been written on this topic and the question is therefore whether the contents and the conclusions of the current manuscript warrant a new installment in this series. In my opinion, the same conclusions have been widely advocated and accepted in the field in recent years. I will further leave it to the authors to explain in their rebuttal letter whether this manuscript contains sufficient novelty over existing ones, such as, for example, Samandari et al (Transl Res 201:98-127, Nov. 2018). The Samandari review is 30 pages long with 3 textbook quality figures, 3 large tables and 280 references (about 40 from 2017 and 2018).

Furthermore, I find that important issues such as sample collection, storage and handling, as well as statistical considerations regarding liquid biopsies have not been covered here that well. The most glaring omission however, is the total absence of any discussion, or even mention thereof, of protein and metabolite blood-based biomarkers. Whereas the use of proteins and metabolites is subject to valid criticisms, they nonetheless have been extensively studied and applied in recent years. Some selected examples are for instance reference 69 of this manuscript, the work from the Hanash lab at MD Anderson (Capello, J Natl Cancer Inst 2017 Apr 1:109(4).  Fahrmann J Natl Cancer Inst 2019 Apr 1; 111(4):372-79.  Capello, Nature Commun 2019 Jan 16;10(1):254), and the use of thrombospondin-2 in combination with CA19-9 from the Zaret lab (Kim, Sci Transl Med 2017, 9(398)).

I recommend the authors revise and amend their manuscript to correct the above-mentioned shortcomings.

Author Response

- We fully agree that many reviews have been previously published on the subject, including that of Samandari et al. in 2018. The latter review, and other before, reported first the technological and molecular aspects of liquid biopsies before giving a summary the clinical interest of such strategies, mainly for the diagnosis of pancreatic cancer.

Quite the contrary, our manuscript opposes the idea of selecting works with clear clinical potential. Accordingly, we only selected papers that i) included both patients with pancreatic cancer and matching controls  (either healthy or with benign pancreatic disease) and ii) that addressed if liquid biopsies could predict or monitor disease progression. In addition, we selected publications including more than 20 (CTC studies) and 50 (ctDNA studies) samples from patients with PDAC. Finally, with regards to the two recently published reviews on the topic (MR. Samandari et al. J Transl Res December 2018, Gall TMH et al. Am J. Pathol January 2019) 10 important publications have been added to our manuscript, including one describing for the first time how ctDNA fragmentation can reveal cancer (lines 271- 279). Noteworthy, the references of these 2 recent reviews have been added in the revised version of the Ms.

- We agree that the problems of standardization of sampling, storage and handling are crucial in order to unify practices and to centralize analysis in expert centres. We discuss this aspect in the last chapter (Chapter 8 – lines 469-471), which is the current role of international consortia.

- The statistical considerations are also important and this aspect is also discussed later in the same Chapter 8 (lines 473-481)

- For circulating proteins and metabolites as a liquid biopsy tool, we have added a new chapter summarizing the main studies recently published in their aims, methods, results and perspectives (new chapter 7) (lines 428-447).

All corrections and additional text appear highlighted in yellow in the revised version.

Round 2

Reviewer 3 Report

Most of my remarks and questions have been satisfactorily addressed and answered by the authers. Three paragraphs on sample handling, statistical analysis, and the use of circulating protein and metabolite biomarkers for PDAC have been added as per my suggestion. The protein / metabolite section is still rather concise compared to the others but at least its existence is acknowledged now and a few pertinent, recent studies referenced. More details about this emerging field can be found in a recent review in CANCERS (2018 Mar 7;10(3). pii: E67. doi: 10.3390/cancers10030067.)

I recommend acceptance of this manuscript.